# Vasopressor-Sparing Strategies in Patients with Shock: A Scoping-Review and an Evidence-Based Strategy Proposition

**DOI:** 10.3390/jcm10143164

**Published:** 2021-07-18

**Authors:** Pierre-Grégoire Guinot, Audrey Martin, Vivien Berthoud, Pierre Voizeux, Loic Bartamian, Erminio Santangelo, Belaid Bouhemad, Maxime Nguyen

**Affiliations:** 1Department of Anesthesiology and Intensive Care, CHU Dijon, 21000 Dijon, France; audrey.martin@chu-dijon.fr (A.M.); viven.berthoud@chu-dijon.fr (V.B.); pierre.voizeux@chu-dijon.fr (P.V.); loic.bartamian@chu-dijon.fr (L.B.); erm.santangelo@gmail.com (E.S.); belaid.bouhemad@chu-dijon.fr (B.B.); maxime.nguyensoenen@gmail.com (M.N.); 2Lipness Team, INSERM Research Center LNC-UMR1231 and LabEx LipSTIC, University of Burgundy, 21000 Dijon, France

**Keywords:** vasopressor, shock, norepinephrine, sepsis, weaning

## Abstract

Despite the abundant literature on vasopressor therapy, few studies have focused on vasopressor-sparing strategies in patients with shock. We performed a scoping-review of the published studies evaluating vasopressor-sparing strategies by analyzing the results from randomized controlled trials conducted in patients with shock, with a focus on vasopressor doses and/or duration reduction. We analyzed 143 studies, mainly performed in septic shock. Our analysis demonstrated that several pharmacological and non-pharmacological strategies are associated with a decrease in the duration of vasopressor therapy. These strategies are as follows: implementing a weaning strategy, vasopressin use, systemic glucocorticoid administration, beta-blockers, and normothermia. On the contrary, early goal directed therapies, including fluid therapy, oral vasopressors, vitamin C, and renal replacement therapy, are not associated with an increase in vasopressor-free days. Based on these results, we proposed an evidence-based vasopressor management strategy.

## 1. Introduction

Because fluid and vasopressors are the main treatments for shock, they are used on a day-to-day basis as symptomatic treatment for arterial hypotension. Vasoplegia is associated with vasodilation and vascular hypo-responsiveness, and involves multiple mechanisms [1]. The aim of vasopressor therapy is to restore organ perfusion so as to limit the risk of multiple organ failure and death. For several years, the published literature regarding the different types of vasopressors (catecholaminergic and non-catecholaminergic agents) has grown considerably. Most of this literature has studied superiority in terms of death or specific clinical outcomes (e.g., acute kidney failure or arrythmia) [2,3,4]. Despite the emergence of new vasopressor agents, norepinephrine is still the recommended first-line agent [5]. One problem with the use of vasopressors is the risk of side effects and the ensuing need for intensive care management, which is costly. Studies have demonstrated that vasopressor use can be associated with specific side effects, and prolonged use may be associated with mortality [6,7]. A study has demonstrated that implementing vasopressor sparing strategies is associated with lower morbidity and ICU (intensive care unit) length of stays [7]. Numerous reviews have investigated the different types of vasopressors and their hemodynamic effects [1,8,9,10,11], but, to date, no review has specifically focused on therapeutic and non-therapeutic strategies associated with the sparing effect for vasopressor use. Thus, we do not have meta-analyses or reviews evaluating the vasopressor-sparing strategies.

Our objective was to perform a scoping-review of the published studies evaluating the vasopressor-sparing strategies. We therefore analyzed the results from randomized controlled trials (RCTs) conducted in patients with shock, focusing on vasopressor doses and/or duration reduction.

## 2. Materials and Methods

### 2.1. Population

We aimed to study vasopressor-sparing strategies in adult patients with shock. We conducted a scoping-review by following PRISMA-scoping review guidelines [12]. Because of its nature, the present scoping-review could not be registered (refusal from Prospero).

### 2.2. Criteria

The inclusion criteria were RCT studies (with parallel groups), with at least one of the reported outcomes including vasopressors. Outcomes were classified as follows: (1) shock reversal, (2) duration of vasopressor use, (3) day free of vasopressors, (4) cumulative dose of vasopressors, and (5) dose of vasopressors.

The exclusion criteria were as follows: articles not available in English, as well as those on traditional Chinese medicine, anaphylactic shock, and reporting COVID-19.

Studies with non-significant outcomes in terms of vasopressor use were reported only if the studies reporting significant outcomes for the same intervention were found.

All of the studies were reviewed by two independent reviewers (A.M. and V.B.). When most studies agreed with the vasopressor-sparing effect, the authors concluded a positive effect. When the results did not meet a consensus, the authors concluded uncertainty, and when most studies were negative, the authors concluded a negative effect.

### 2.3. Algorithm and Study Selection

We included all of the relevant studies found in the Medline database (Pubmed) and Cochrane Library from searches conducted in May 2020 and updated in March 2021, using the following algorithm: (“Cardiovascular Agents/therapeutic use” [Mesh Terms] OR “Cardiovascular Agents/administration and dosage” [Mesh Terms] OR “vasoconstrictor agents/administration and dosage” [Mesh Terms] OR “norepinephrine/administration and dosage” [Mesh Terms] OR “Catecholamines/administration and dosage” [Mesh Terms] OR “Vasoconstrictor Agents/therapeutic use” OR “Catecholamines/therapeutic use” [Mesh Terms] OR “norepinephrine/therapeutic use” [MeSH Terms] OR “Shock/therapy” [MeSH Terms] OR “Algorithms” [MeSH Terms]) AND “humans” [MeSH Terms] AND (“adult” [MeSH Terms] OR “aged” [MeSH Terms] OR “middle aged” [MeSH Terms]) AND (“Shock” [MeSH Terms] OR “Multiple Organ Failure” [MeSH Terms] OR (“Lactic Acid/blood” [MeSH Terms] AND (“vasoconstrictor agents/administration and dosage” [Mesh Terms] OR “norepinephrine/administration and dosage” [Mesh Terms] OR “Catecholamines/administration and dosage” [Mesh Terms] OR “Vasoconstrictor Agents/therapeutic use” OR “Catecholamines/therapeutic use” [Mesh Terms] OR “norepinephrine/therapeutic use” [MeSH Terms] “Cardiovascular Agents/therapeutic use” [Mesh Terms] OR “Cardiovascular Agents/administration and dosage” [Mesh Terms])) AND Randomized Controlled Trial [Publication Type]). The search included all publications since 1995.

## 3. Results

Among the 830 studies screened, 143 were included (Figure 1). Forty studies were not reported because the intervention was only negative for the intervention presented. The main reason for vasopressor administration was septic shock (75%); 6% of studies included patients with cardiogenic shock. Other reason for vasopressor administrations were post cardiopulmonary bypass (3%), post-operative (2%), post cardiac arrest (1%), burn (3%), hypovolemia (1%), distributive shock (1%), vasodilatory shock (2%), and trauma (1%). In 4% of studies, the causes of shock were multiple. Only four RCTs were specifically related to vasopressor weaning [7,13,14,15]. Three studies reported therapeutic algorithms and one study reported a pharmacological intervention.

### 3.1. Pharmacological Interventions

The main results are summarized in Table 1.

#### 3.1.1. Vasopressor

The SOAP (Sepsis Occurrence in Acutely Ill Patients) II study, which compared norepinephrine to dopamine, demonstrated a higher number of vasopressor-free days with norepinephrine [3]. In a small trial, norepinephrine doses were lower when randomized against phenylephrine [94]. Similar vasopressor outcomes were observed with epinephrine compared with norepinephrine [95,96] and with norepinephrine plus dobutamine [97,98]. Many RCTs have compared therapeutic strategies by combining a catecholaminergic to a non-catecholaminergic vasopressor. These RCTs mainly demonstrated that adding a non-catecholaminergic vasopressor decreased the catecholamine dose. The non-catecholaminergic vasopressors studied were vasopressin [20,24,25,26,27,28,29] and its derivatives (terlipressin [18,19,20,23], selepressin [99]), as well as angiotensin-2 [32]. The effect of vasopressin seems to be dose-dependent [100]. Two small studies did not demonstrate differences in catecholamine administration in patients treated with terlipressin [21] and angiotensin-2 [33]. Vasopressin has also been studied in several large RCTs. The VANCS (Vasopressin Versus Norepinephrine for the Management of Shock After Cardiac Surgery) study demonstrated a shorter duration of vasopressor use [30]. The VANISH (Vasopressin vs. Norepinephrine as Initial Therapy in Septic Shock) and the VASST trial found no difference in vasopressor duration in patients with septic shock [4,31], but in the VASST (Vasopressin and Septic Shock Trial) study, vasopressin was associated with a lower dose of norepinephrine. In 526 patients with septic shock, the terlipressin and norepinephrine duration did not differ [22]. Another study comparing dopamine and terlipressin did not demonstrate any difference in vasopressor-free days [101].

#### 3.1.2. Adjuvant

The most studied adjuvant therapy was glucocorticoids [34,35,36,37,38,39,40,41,42,45,46,47,48,49,50,51,52]. In large RCTs, substitutive corticotherapy consistently decreased the time to shock reversal [38,39,40,41]. Patients were mostly included in the early phase of septic shock. One RCT demonstrated an improvement in shock reversal in patients with late septic shock (i.e., >48 h) [36]. Hydrocortisone as an adjunctive treatment of vasopressin also decreased the duration of vasopressor administration [42]. Only one trial evaluated mineralocorticoid against a placebo. In this 2 × 2 factorial trial, fludrocortisone alone was not associated with more vasopressor-free days than the placebo [72].

Vitamin C has also been the object of several negative RCTs [76,77,78]. Two RCTs demonstrated a decrease in the time to resolution of shock in patients treated with a combination of thiamine, hydrocortisone, and vitamin C. However, because a combination of several drugs, including hydrocortisone, was used, the effect of vitamin C was not evaluated [43,44].

In patients with septic shock and tachycardia, esmolol decreased the norepinephrine requirements [69]. In 70 patients with multi-organ dysfunction, a heart rate higher than 90 BPM (beats per minute), and a contraindication to beta-blockers, ivabradine administration did not decrease vasopressor use [102].

Two small randomized trials evaluating methylene blue demonstrated a decrease in vasopressor use in both septic and post cardiopulmonary bypass vasoplegia [70,71]. *N*-acetyl cysteine has been reported to decrease a composite vaso-inotropic score in burn patients [75]. However, two studies performed in septic shock did not confirm these results [73,74].

Drotrecogin alfa (activated) [103], pyridoxalated hemoglobin polyoxyethylene (PHP) [104,105], nitric oxyide synthase inhibitors [106,107,108], and monoclonal antibody to human tumor necrosis factor [109,110] have been associated with decreased norepinephrine administration, but these drugs were abandoned for safety reasons (PHP [104] and nitric oxide synthase inhibitor [106]) or lack of efficacy (monoclonal antibody to human tumor necrosis factor [111], and Drotrecogin alfa [103,112,113]).

One RCT study demonstrated a decrease in norepinephrine and dopamine administration with heptaminol [15]. Midodrine is another oral vasoconstrictor frequently administered to reduce the time to norepinephrine weaning. Midodrine has been inconsistently associated with shorter vasopressor administration and ICU length of stay [114,115,116]. The recent MIDAS (effect of midodrine versus placebo on time to vasopressor discontinuation in patients with persistent hypotension in the intensive care unit) trial reported that midodrine did not reduce time to vasopressor discontinuation in patients with persistent hypotension [79].

### 3.2. Fluid Therapy

During early goal-directed therapy (EGDT), the relationship between the volume of fluid administered and the norepinephrine doses were inconstant [53,54,57,58,59,60,62,63]. One RCT evaluating a restrictive fluid strategy did not result in increased doses of vasopressors, despite the higher volumes of administered fluid [61], whereas a second study found no differences in fluid volume and vasopressor use between groups [56]. Fluid titration based on dynamic preload parameters resulted in a similar shock duration, despite a lower fluid intake [64]. In line with this result, two RCTs evaluating a hemodynamic strategy based on cardiac output monitoring did not demonstrate a decrease in vasopressor treatment [65,66]. In another trial, PICCO (Pulse Contour Cardiac Output) guided resuscitation was associated with the administration of fewer vasopressors [55]. The CRISTAL-RCT (Colloids Versus Crystalloids for the Resuscitation of the Critically Ill) reported more days alive without vasopressors with colloid use in comparison with crystalloid use [67]. In the ALBIOS (Albumin Italian Outcome Sepsis) study, the administration of albumin decreased both the time to vasopressor or inotropic agent cessation and the fluid balance [68].

### 3.3. Body Temperature

In septic patients, external fever control aiming for 36.5 to 37 °C decreased the vasopressor requirement [80]. Hypothermia (32–34 °C) was associated with fewer vasopressor-free days [81]. In patients with cardiogenic shock, moderate hypothermia (33 °C) did not decrease vasopressor administration [82].

### 3.4. Kidney Replacement Therapy

In patients with septic shock, the early application of continuous veno-venous filtration resulted in a longer time to shock reversal [117]. Three small studies on high volume hemofiltration [83,84] and cytosorb therapy [89] demonstrated a decrease in vasopressor treatment. Several studies including high volume hemofiltration did not confirm those results [85,86,87,88]. Adsorption did not allow for a decrease in norepinephrine administration [90,91,92,93]. One study comparing adsorption to high volume hemofiltration did not demonstrate any differences [118].

## 4. Discussion

This review suggests that several pharmacological and non-pharmacological strategies are associated with a decrease in the duration of vasopressor therapy. These strategies are as follows. (1) The implementation of a blood pressure objective affects vasopressor duration, but without evidence in terms of mortality. (2) Implementing a weaning strategy decreases the duration of vasopressor treatment and ICU stays. (3) Vasopressin may decrease norepinephrine doses—weaning norepinephrine after vasopressin in case of co-administration seemed to decrease hypotensive episodes. (4) Systemic glucocorticoid administration increased the number of vasopressor-free days. (5) Beta-blockers might decrease norepinephrine doses and duration in selected patients. (6) Targeting normothermia might make it feasible to decrease vasopressor administration. (7) On the contrary, an analysis of the literature demonstrated that EGDT, including fluid therapy, oral vasopressors, vitamin C, and renal replacement therapy, are not associated with an increase in vasopressor-free days.

### 4.1. Pharmacological Strategies

Adding a non-catecholaminergic vasopressor to norepinephrine makes is possible to lower its dosage, and the administration of a non-catecholaminergic drug is recommended for patients in refractory shock [5]. Vasopressin deficiency is part of the pathological mechanisms leading to vasoplegia in septic shock [119], and might explain the lower tolerance to vasopressin weaning compared with norepinephrine [14]. However, meta-analysis did not report any differences in mortality or length of stay [120,121]. Angiotensin-2 is likely to have similar effects, but has been less studied because it only recently received FDA (Food and Drug Administration) approval. Thus, in a context where vasopressin and norepinephrine are being administered, it seems that norepinephrine should be weaned first [14,121]. However, because of its inotropic effect, norepinephrine should be weaned depending on the myocardial contractility.

Glucocorticoids are known to restore vascular responsiveness [122], and can be used to treat corticosteroid insufficiency [123] and block the synthesis of pro-inflammatory cytokines [124]. Substitutive corticotherapy was the most described adjuvant treatment, and it was reported to consistently decrease time to shock reversal. However, the timing of this intervention was not consistent in every trial, and in the most recent surviving sepsis guidelines, glucocorticoids are only advised in refractory shock [5]. One trial specifically focused on late shock reversal demonstrated that low-dose glucocorticoids are beneficial in vasopressor weaning, mainly in patients with sepsis and a high dose of vasopressors (more than 0.20 µg/kg/min).

Cardio-selective beta-blockers were demonstrated to lower vasopressor doses in patients with sepsis and tachycardia [69]. This phenomenon was attributed to a decrease in arterial load associated with an improvement in ventriculo-arterial coupling [125]. During weaning, the rate control might improve the ventriculo-arterial coupling [126], suggesting that it could be an interesting therapeutic approach in selected patients.

During hemodynamic resuscitation, fluid therapy does not result in shorter vasopressor treatment, and could even increase vasopressor duration [127]. Thus, it is unlikely that uncontrolled administration of fluids during the optimization and weaning phases would reduce vasopressor duration. Because increasing vasopressors might increase cardiac preload [128,129], in the context of weaning, hypotension might be related to preload reduction. Because removing fluids is another cornerstone of de-resuscitation [130], we believe that preload status should be carefully evaluated in patients with hypotension attributed to vasopressor weaning.

### 4.2. Hemodynamic Strategies

In practice, norepinephrine weaning is implemented empirically: when the arterial pressure is consistently above the given objective, the dose of norepinephrine is decreased until discontinuation. Because vasopressors act on several pathological mechanisms involved in blood pressure (preload [128], inotropism [10], and vascular resistance [10]), arterial hypotension following a decrease in vasopressor dose can have multiple causes, and several studies demonstrated that arterial dynamic elastance (E_Adyn_ = respiratory pulse pressure variation/respiratory stroke volume variation), which may reflect ventriculo-arterial coupling and vasomotor tone, can predict the pressure response to norepinephrine weaning [131]. An E_Adyn_-based algorithm decreased norepinephrine administration [7], and the automation of the weaning process by a controller responding to “fuzzy logic” made it possible to reduce the weaning time [13]. Because targeting the lower blood pressure target is associated with faster vasopressor discontinuation, it is clear that setting the right pressure target is an important decision. During acute circulatory failure, a minimal threshold for blood pressure is often set in order to ensure organ perfusion. In a recent study, setting a low blood pressure target (60 to 65 mmHg) in patients older than 65 years old with vasodilatory hypotension resulted in lower exposure to vasopressors, without a significant difference in morbidity or mortality [132]. This trial further suggested that lower blood pressure might be targeted in a selected resuscitated population, and that a lower objective might result in lower vasopressor exposure without adverse events. Nevertheless, blood pressure targets should be individualized to ensure organ perfusion [133].

### 4.3. Proposed Algorithm

Because vasopressors are associated with adverse effects [134], vasopressor-sparing strategies have emerged [135]. Vasopressor sparing strategies aim to reduce vasopressor exposure and their side effects [7]. Vasopressor administration is part of the continuum from initial resuscitation to shock reversal. As for fluid therapy, the management of vasopressors can be divided into several phases: an initial resuscitation phase with the objective of obtaining the target blood pressure to restore tissue perfusion as quickly as possible, a stabilization phase, and a weaning phase (Figure 2). When the patient is stabilized, active management, including pharmacological therapies and hemodynamic strategies, may be introduced to decrease the time to vasopressor discontinuation. Hemodynamic stability can be defined as a blood pressure variation of less than 10% without changing the vasopressor dose, and with an improvement in tissue perfusion (capillary refill time, hyperlactatemia, diuresis, and venous oxygen saturation) [7]. When blood pressure is controlled, the cardiac and vascular properties are gradually restored and the withdrawal of vasopressor drugs can be initiated (Figure 2). As the pathological process may not be symmetrical during acute circulatory failure and recovery, and as catecholamine administration leads to the down-regulation of adrenergic receptors [30], it is unlikely that vasopressor withdrawal is a parallel process to the initial resuscitation phase.

Optimization and weaning should be part of a protocolized process (Figure 2). Firstly, a blood pressure target aiming to optimize organ perfusion should be defined by taking into account that a lower blood pressure objective may be possible. Secondly, physicians should consider the early use of intravenous hydrocortisone, non-catecholaminergic vasopressors (vasopressin and blue methylene), and maintaining normothermia. In selected patients with persistent tachycardia, beta-blocker therapy should be considered. Thirdly, when hemodynamic stability is obtained, active weaning should be initiated following a hemodynamic algorithm (see below). Because of their action on preload, inotropism, and arterial load, vasopressor withdrawal should be adapted to the patient’s haemodynamic profile, similarly to therapeutic escalation in the acute phase.

Several approaches can be followed depending on the type of monitoring (Figure 3). Basic management based on continuous measurement of blood pressure is probably the most commonly used approach. In the case of poor tolerance, the different causes listed above should be considered and eliminated in the order of the following frequency: preload, arterial load, and inotropism. A hemodynamic approach based on the analysis of ventriculo-arterial coupling and/or E_Adyn_ may be suitable [131,136,137,138,139]. Studies have demonstrated that E_Adyn_ and the analysis of ventriculo-arterial coupling are able to predict the effects of norepinephrine on stroke volume and blood pressure [7,140]. Arterial elastance (E_A_) is an index of arterial load [141] that integrates the main components of arterial load (i.e., total peripheral resistance, total net arterial compliance, characteristic impedance, and systolic and diastolic time intervals). E_A_ can be estimated by using the equation E_A_ = mean arterial pressure/stroke volume (mmHg/mL) [142]. Left ventricular end-systolic elastance (E_V_) is an indicator of cardiac function [143], and might be extrapolated by using the non-invasive single beat method described by Chen et al. [144]. Ventriculo-arterial coupling is the ratio of E_A_ to E_V_, and it is an indicator of the balance between cardiac effort and arterial load. When the left ventricle and the vascular system are coupled, this ratio is around 1 [145]. Such approaches are made possible by the use of echocardiography or a continuous hemodynamic monitoring system [136,137,138,139]. In the future, physicians could consider using automation and artificial intelligence to guide the weaning process [13,146]. A recent study published in the JAMA showed that the use of this index with a therapeutic management decision algorithm reduces hypotension time [146].

Several limitations can be discussed. We can question the clinical benefit of a vasopressor sparing strategy. There is no direct evidence that reducing pressor “stress” decreases mortality. However, studies have demonstrated that decreasing the duration of vasopressor use is associated with a decrease of ICU length of stays and morbidity [7]. Moreover, a pooled analysis of two studies suggested that increased exposure to vasopressor increased the risk of death [6]. Because vasopressor adverse effect are largely documented, sparing vasopressor and/or catecholamine is a topic of growing interest [135,147]. Most included studies were performed in sepsis (75%) or postoperative vasoplegic shock, thus limiting the extrapolation of our scoping review to these types of shock. According to the PRISMA-scr guidelines, the critical appraisal of the included sources of evidence was not mandatory. Because of the study design, the clinical benefit of the reviewed interventions was determined by the authors, and the criteria may be subjective.

## 5. Conclusions

There are a few published RCTs focused specifically on sparing strategies. In patients with shock, several pharmacological strategies, such as hydrocortisone, can be safely used to optimize vasopressor treatment. In addition, it appears that optimized patient management using hemodynamic guidelines could be associated with more vasopressor-free days and shorter ICU stays.

## Figures and Tables

**Figure 1 jcm-10-03164-f001:**
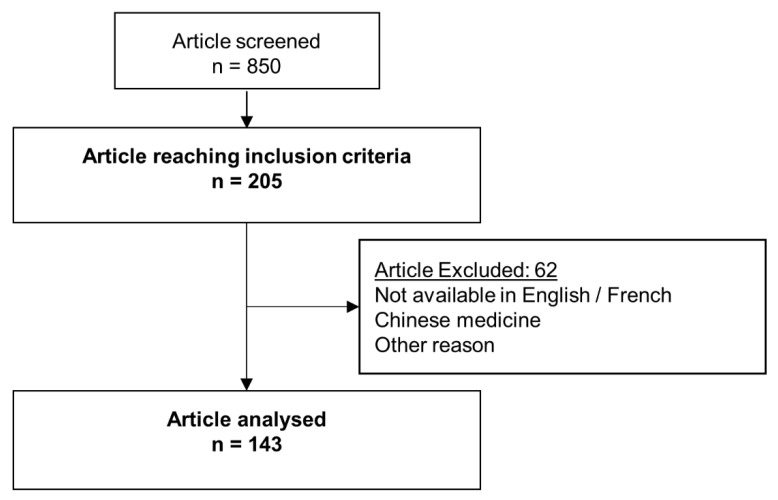
Flow chart study.

**Figure 2 jcm-10-03164-f002:**
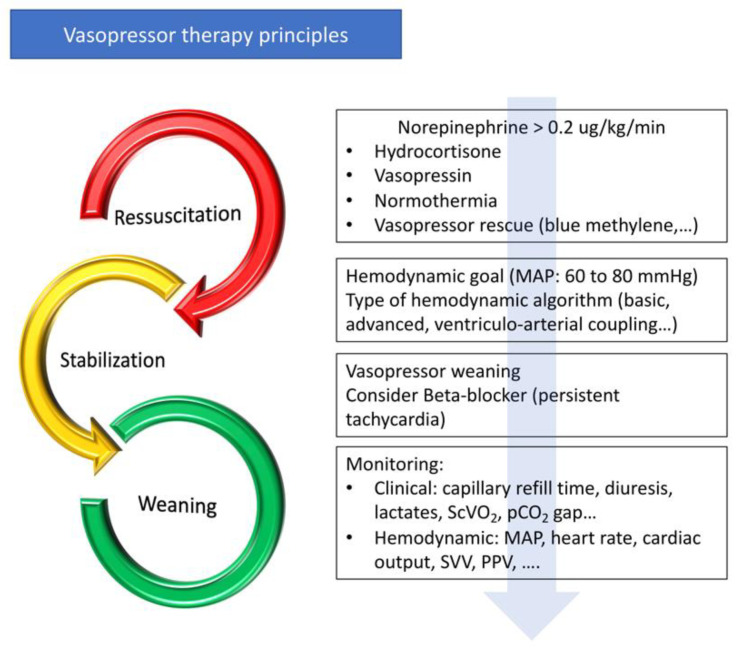
Principles of vasopressor treatment. MAP—mean arterial pressure; ScVO_2_—central venous oxygen saturation; pCO_2_ Gap—difference between partial pressure of CO_2_ in venous blood and arterial blood; SVV—stroke volume variation; PPV—pulse pressure variation.

**Figure 3 jcm-10-03164-f003:**
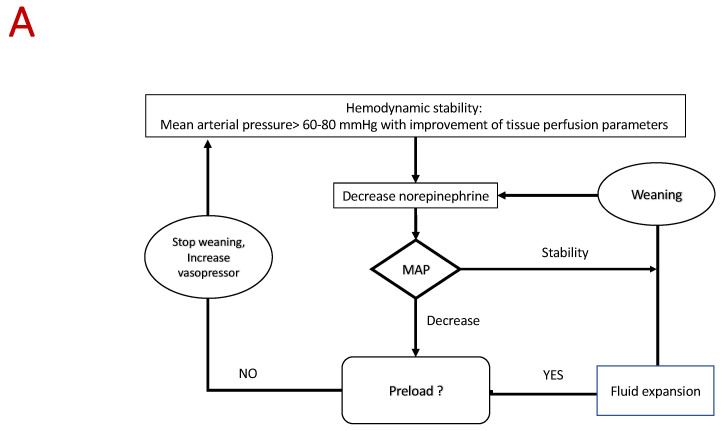
Algorithm proposed for vasopressor weaning. (**A**) Basic algorithm. (**B**) Advanced algorithm based on dynamic arterial elastance analysis. (**C**) Advanced algorithm based on ventriculo-arterial coupling analysis. E_Adyn_—dynamic arterial elastance; PPV—pulse pressure variation; SVV—stroke volume variation; E_v_—ventricular elastance; E_A_—arterial elastance.

**Table 1 jcm-10-03164-t001:** Main intervention and proposed use.

Intervention	References	Outcome Evaluated	Clinical Effect	Proposal
**Hemodynamic management**				
Active, hemodynamic algorithm	[7,13]	Dose and duration	Positive	Systematic
Blood pressure target	[16,17]	Dose and duration	Positive	Individualized on organ perfusion
Wean norepinephrine before vasopressin	[14]	Duration	Uncertain	Wean norepinephrine first
**Pharmocological**				
Dopamine	[3]	Dose and duration	Negative	No
Terlipressin	[18,19,20,21,22,23]	Dose and duration	Negative	No
Vasopressin	[4,20,24,25,26,27,28,29,30,31]	Dose and duration	Uncertain	Consider if norepinephrine > 0.2 µg/kg/min
Angiotensin-2	[32,33]	Dose and duration	Uncertain	Mores studies need
Glucocorticoids	[34,35,36,37,38,39,40,41,42,43,44,45,46,47,48,49,50,51,52]	Dose and duration	Positive	Systematic if norepinephrine > 0.2 µg/kg/min
Fluid therapy and hemodynamic goal direct therapy	[53,54,55,56,57,58,59,60,61,62,63,64,65,66]	Dose and duration	Uncertain	Assess preload dependency
Colloid	[67,68]	Dose and duration	Positive	Not recommended
Beta-blockers	[69]	Dose	Uncertain	Selected population with persistent tachycardia
Methylene blue	[70,71]	Dose and duration	Uncertain	Rescue
Mineralocorticosteroids	[39,72]	Duration	Negative	No
*N*-acetyl cysteine	[73,74,75]	Dose and duration	Uncertain	No
Vitamin C	[43,44,73,76,77,78]	Dose and duration	Negative	No
Oral vasopressor	[15,79]	Dose and duration	Negative	No
**Non-pharmacological**				
Body temperature	[80,81,82]	Dose and duration	Positive	Target normothermia
High volume hemo-filtration	[83,84,85,86,87,88]	Dose and duration	Negative	No
Adsorption	[89,90,91,92,93]	Dose and duration	Uncertain	No

## Data Availability

The data presented in the study are available upon request from the corresponding author.

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
