# Peer review of "Vasopressor-Sparing Strategies in Patients with Shock: A Scoping-Review and an Evidence-Based Strategy Proposition"

_jcm, 2021, doi:10.3390/jcm10143164_

Round 1

Reviewer 1 Report

The authors adress the very and original topic of vasopressor sparing. They describe in a structured way several methods to spare vasopressors. The message they want to bring is - in my opinion - what other methods/therapies can spare the use of vasopressors? I really miss if sparing of vasopressors may reduce mortality and morbidity. 

Major comments:

I miss a thorough discussion on benefit of vasopressor sparing. Is there real evidence that this will save lifes or reduce length of stay?

Please provide also evidence for lower target blood pressures in post-shock patients weaning from vasopressors.

45: "conducted between May 2020 and March 2021" Why such a short period of time? Why did you not use the term "vasopressor" or vasopressor sparing therapies"?

70: molecule??

184: "Thus, in a context where multiple vasopressors are being administered, it seems that norepinephrine should be weaned first." You should provide more proof for this recommendation. Is the outcome with norepinephrine really worse compared to other drugs? Are there studies comparing this? 

199: rhythm control; do you mean "rate control"?

204: "Seeing as vasopressors might increase cardiac preload [121,122], weaning might result in hypotension related to preload reduction. What do you mean here?

minor comments:

Improve English writing and style considerably: e.g.:

2: Vasopressor-sparing Strategies in patients with shock: 

10: vasopressor therapy

11: we performed

13: with a focus on on reduction of vasopressor doses and duration

14: our analysis

Author Response

Reviewer 1

Comments and Suggestions for Authors

The authors adress the very and original topic of vasopressor sparing. They describe in a structured way several methods to spare vasopressors. The message they want to bring is - in my opinion - what other methods/therapies can spare the use of vasopressors? I really miss if sparing of vasopressors may reduce mortality and morbidity. 

Response:

We thank the reviewer for his careful reading of our manuscript, and his helpful comments. As explained in introduction section, few studies have focused on this topic whereas vasopressor use is associated with side effects. For example, the VANCS study has demonstrated lower incidence of renal failure or arrythmia with the use of vasopressin and norepinephrine vs norepinephrine alone. A pooled analyze of two RCT studies demonstrated that increased exposure to vasopressor increased the risk of death. A randomized study on vasopressor sparing strategies has demonstrated lower incidence of arrythmia and decrease ICU length of stay. The main objective of the present study was not to demonstrated that vasopressor sparing strategies decrease mortality. On contrary, the objective was to perform a scoping review to propose strategies, and further direction on research.

Major comments:

I miss a thorough discussion on benefit of vasopressor sparing. Is there real evidence that this will save lifes or reduce length of stay?

We thank the reviewer for his relevant comment. Because vasopressor adverse effects are largely documented, it seems intuitive that reducing exposure to vasopressor would decrease their adverse effects. Thus, sparing vasopressor / catecholamine  is a topic of growing interest (as for fluid therapy). However, even though we agree with the reviewer that there is no direct evidence that reducing vasopressor decrease mortality, the results from an  analysis based on SEPSISPAM and OVATION trial suggest an increased risk of death in patients treated longer than 6 hours. We discussed these points in the introduction and limitation sections.

Please provide also evidence for lower target blood pressures in post-shock patients weaning from vasopressors.

We thank the reviewer for his comment. The fact that lower blood pressure might be tolerated in post-shock patients has recently been documented in a RCT conducted in  patients older than 65 years old (Lamontagne F, Richards-Belle A, Thomas K, Harrison DA, Sadique MZ, Grieve RD, Camsooksai J, Darnell R, Gordon AC, Henry D, Hudson N, Mason AJ, Saull M, Whitman C, Young JD, Rowan KM, Mouncey PR. Effect of Reduced Exposure to Vasopressors on 90-Day Mortality in Older Critically Ill Patients with Vasodilatory Hypotension: A Randomized Clinical Trial. JAMA - J Am Med Assoc. 2020. doi:10.1001/jama.2020.0930.). (line 228). Equally, the sepsis-PAM study (Asfar P, Meziani F, Hamel JF, Grelon F, Megarbane B, Anguel N, Mira JP, Dequin PF, Gergaud S, Weiss N, Legay F, Le Tulzo Y, Conrad M, Robert R, Gonzalez F, Guitton C, Tamion F, Tonnelier JM, Guezennec P, Van Der Linden T, Vieillard-Baron A, Mariotte E, Pradel G, Lesieur O, Ricard JD, Hervé F, du Cheyron D, Guerin C, Mercat A, Teboul JL, Radermacher P; SEPSISPAM Investigators. High versus low blood-pressure target in patients with septic shock. N Engl J Med. 2014 Apr 24;370(17):1583-93. doi: 10.1056/NEJMoa1312173. Epub 2014 Mar 18. PMID: 24635770) did not demonstrate any clinical benefit of high blood pressure over low blood pressure in septic shock. We added these points in the discussion section.

45: "conducted between May 2020 and March 2021" Why such a short period of time? Why did you not use the term "vasopressor" or vasopressor sparing therapies"?

We thank the reviewer for his comment. We apologize for the lack of clarity. The original review has been conducted on may 2020 and updated in march 2021. This has been corrected in the method section (line 48)

70: molecule??

We apologize for the lack of clarity. The mistake has been clarified.

184: "Thus, in a context where multiple vasopressors are being administered, it seems that norepinephrine should be weaned first." You should provide more proof for this recommendation. Is the outcome with norepinephrine really worse compared to other drugs? Are there studies comparing this? 

We thank the reviewer for his comment. We apologize for the lack of clarity. Several studies and one meta-analysis demonstrated that weaning NE first was associated with higher incidence of hypotension (Wu Z, Zhang S, Xu J, Xie J, Huang L, Huang Y, Yang Y, Qiu H. Norepinephrine vs Vasopressin: Which Vasopressor Should Be Discontinued First in Septic Shock? A Meta-Analysis. Shock. 2020 Jan;53(1):50-57. doi: 10.1097/SHK.0000000000001345. PMID: 31008869). These points have been clarified in the discussion section (line 183-186).

199: rhythm control; do you mean "rate control"?

We thank the reviewer for his relevant comment. The mistake has been corrected

204: "Seeing as vasopressors might increase cardiac preload [121,122], weaning might result in hypotension related to preload reduction. What do you mean here?

We thank the reviewer for his comment. We apologize for the lack of clarity, the sentence has been clarified

minor comments:

Improve English writing and style considerably: e.g.:

We thank the reviewer for his comment. The manuscript has been reviewed by an english native speaker.

2: Vasopressor-sparing Strategies in patients with shock: 

10: vasopressor therapy

11: we performed

13: with a focus on on reduction of vasopressor doses and duration

14: our analysis

We thank the reviewer for his comment. We apologize for those spelling mistake that have been corrected

Reviewer 2 Report

The authors performed a review about the vasopressors sparing strategy. Although the topic may be of interest, there are several limitations:

  • The paper is not fluid to read.
  • The methodology used is not clear. The paper has some characteristics of systematic review but this approach is not clearly declared in the title and methods section.
  • The PICO questions are not clearly defined. Particularly, the population and clinical condition of interest are not adequately described (adult and/or pediatric population?, any type of shock?).
  • The outcome n.6 (row 67) does not seem appropriate. It is an exclusion criteria.
  • The assessment of studies' quality has not been done.
  • The results are not clear and exhaustively reported. Different types of shock are not mentioned.
  • Table 1 is not clear. Criteria to determine "clinical beneficence" are not reported.
  • Row 215: the authors defined Eady as "respiratory stroke volume variation / respiratory pulse pressure variation". However, Eady is PPV/SVV.
  • Figure 3B should report Eadyn<1 instead of SVV>PPV, and Eadyn>1 instead of SVV<PPV.
  • Row 280-281: reference needed.

Author Response

Reviewer 2

The authors performed a review about the vasopressors sparing strategy. Although the topic may be of interest, there are several limitations:

  • The paper is not fluid to read. The methodology used is not clear. The paper has some characteristics of systematic review but this approach is not clearly declared in the title and methods section.

We thank the reviewer for his helpful comment. We would like to apologize for the lack of clarity that may be due to the design of this review. As explained in the introduction section, the literature on vasopressor sparing strategies is sparse. We have several published studies that can provide information on duration of vasopressor therapy. Because we aimed to provide an overview of vasopressor-sparing strategies and not to provide an answer to a binary question, we conducted a scoping review. This scoping review follows PRISMA guidelines on scoping review. The term scoping review has been added in the title, the abstract and the method sections.

The PICO questions are not clearly defined. Particularly, the population and clinical condition of interest are not adequately described (adult and/or pediatric population?, any type of shock?).

We thank the reviewer for his comment. We apologize for the lack of clarity. As pointed out by the reviewer, the population of interest was adult patients with shock. We clarified this point in the method section.

The outcome n.6 (row 67) does not seem appropriate. It is an exclusion criteria.

We thank the reviewer for his comment. The outcome n.6 has been added as an exclusion criteria

The assessment of studies' quality has not been done.

We thank the reviewer for his comment. Because of its design (i.e scoping review), the studie’s quality assessment was not mandatory. Nevertheless, we agree with the reviewer that may be a limitation. This fact has been discussed in the limitation section.

The results are not clear and exhaustively reported. Different types of shock are not mentioned.

We thank the reviewer for his comment. The design of this scoping review follows PRISMA-scr guidelines. Because the subject is wide with several studies, we report the results as pharmalogical and non-pharmalogical interventions. Then we followed this plan for the results and discussion section. We added the type of shock has in the result section (line 81-84).

Table 1 is not clear. Criteria to determine "clinical beneficence" are not reported.

We thank the reviewer for his comment. We changed the term by “clinical effect” according to the demonstrated effect. Thus, the clinical effect was determined by the authors based on the evidence previously reported.  When most studies agreed with the effect, we concluded to the positive effect. When results were not consensual, we concluded to uncertain and when most studies were negative, we concluded to negative effect. Because we performed a scoping review and not a meta-analysis, this criterion is not objective and could be subject to discussion. We discussed this fact in the limitation section.

Row 215: the authors defined Eady as "respiratory stroke volume variation / respiratory pulse pressure variation". However, Eady is PPV/SVV.

We thank the reviewer for his comment. We made the change.

Figure 3B should report Eadyn<1 instead of SVV>PPV, and Eadyn>1 instead of SVV<PPV.

We thank the reviewer for his comment. We made the change.

Row 280-281: reference needed.

We thank the reviewer for his comment. The sentence has been removed in order to improve clarity.

Reviewer 3 Report

The article "Vasopressor-sparing strategies in shocked patients: A scoping review and evidence-based strategy proposition" reviews recent RCTs and proposes a well-designed strategy to save the doses of vasopressors for avoiding adverse effects. The work is quite well and suitable to provide updated information. I suggest some issues before acceptance. 

Major problems

1. The author proposed an algorithm for sparing vasopressor in Figures 2 and 3. However, as the author mentioned in the methods, the evidence was built by septic shock. Strategy to manage hypovolemic shock could be intrinsically different from distributive shock, and even permissive hypotension could be available to prevent additional bleeding. Furthermore, the early use of hydrocortisone had not been sufficiently proven in hypovolemic or obstructive shock. Therefore, the author needs to discuss more detail about these in the discussion section.

Minor problems

1. The author mentioned previous studies without adequate references. Inserting references after these sentences would help readers.

"Numerous reviews have investigated the different types of vasopressors and their hemodynamic effects," (line 34-35)

"Because vasopressors act on several pathological mechanisms involved in blood pressure, arterial hypotension following a decrease in vasopressor dose can have multiple causes, ... " (line 212-217)

"A recent study published in the JAMA has shown that the use of this index with a therapeutic management decision algorithm reduces hypotension time." (line 280-282)

2. In Table 1, the clinical benefice of the vasopressin was missing. Please insert it.

3. In line 113, delete double space between words.  

"including hydrocortisone  was" -> "including hydrocortisone was"

4. In line 127, use full-term before using the abbreviation "PHP".

5. In Figure 3, the figure subtitles of B ("Algorithm proposed for vasopressor weaning." and  C ("Algorithm proposed for vasopressor weaning.")should be separate from the legend.

6. Consider change Figure 3 (B and C) of the algorithm to enhance understanding. e.g., SVV > PPV -> yes or no -> increase preload, vasopressor or beta-blocker, inotrope.

Author Response

Reviewer 3

The article "Vasopressor-sparing strategies in shocked patients: A scoping review and evidence-based strategy proposition" reviews recent RCTs and proposes a well-designed strategy to save the doses of vasopressors for avoiding adverse effects. The work is quite well and suitable to provide updated information. I suggest some issues before acceptance. 

Major problems

  1. The author proposed an algorithm for sparing vasopressor in Figures 2 and 3. However, as the author mentioned in the methods, the evidence was built by septic shock. Strategy to manage hypovolemic shock could be intrinsically different from distributive shock, and even permissive hypotension could be available to prevent additional bleeding. Furthermore, the early use of hydrocortisone had not been sufficiently proven in hypovolemic or obstructive shock. Therefore, the author needs to discuss more detail about these in the discussion section.

We thank the reviewer for his relevant comment. We agree that most evidence are based on sepsis and postoperative area, the interpretation of our findings in hypovolemic shock may be limited. But most of patients admitted to ICU suffers of septic or perioperative vasoplegia. Despite this point, we believe that our scoping review fit with daily vasoplegic syndrom. We discussed this point in the limitation section.

Minor problems

  1. The author mentioned previous studies without adequate references. Inserting references after these sentences would help readers.

"Numerous reviews have investigated the different types of vasopressors and their hemodynamic effects," (line 34-35)

"Because vasopressors act on several pathological mechanisms involved in blood pressure, arterial hypotension following a decrease in vasopressor dose can have multiple causes, ... " (line 212-217)

"A recent study published in the JAMA has shown that the use of this index with a therapeutic management decision algorithm reduces hypotension time." (line 280-282)

 We thank the reviewer for his relevant comment. We apologize for the lack of clarity, and we added references.

  1. In Table 1, the clinical benefice of the vasopressin was missing. Please insert it.

We thank the reviewer for his comment. We made the change.

  1. In line 113, delete double space between words.  

"including hydrocortisone was" -> "including hydrocortisone was"

We thank the reviewer for his comment. We made the change.

  1. In line 127, use full-term before using the abbreviation "PHP".

We thank the reviewer for his comment. We made the change.

  1. In Figure 3, the figure subtitles of B ("Algorithm proposed for vasopressor weaning." and  C ("Algorithm proposed for vasopressor weaning.")should be separate from the legend.

We thank the reviewer for his comment. We made the change.

  1. Consider change Figure 3 (B and C) of the algorithm to enhance understanding. e.g., SVV > PPV -> yes or no -> increase preload, vasopressor or beta-blocker, inotrope.

We thank the reviewer for his comment. We made the change, and SVV> PPV has been replaced by Eadyn > 1 as requested by another reviewer.

Round 2

Reviewer 1 Report

Thanks for your corrections

Author Response

We thank the reviewer for his comment.

Reviewer 2 Report

The authors have clarified that they follow the methodology of scoping review. However, some concerns are still present that limit the rigor of the manuscript. The target population is wide, including all patients with shock. Patients with cardiogenic shock are excluded (row 62) even if they are an important category that may benefit from vasopressor weaning. Despite this, the authors stated that 6% of the included studies considered patients with cardiogenic shock (rows 92-93).

Author Response

We thank the reviewer for his relevant comments. We did not exclude cardiogenic shock. As written in the first version of the submitted manuscript: “Exclusion criteria were as follows: article not available in English, and traditional Chinese medicine. Studies with non-significant outcomes in term of vasopressor use were presented only if studies reporting significant outcomes for the same intervention/ molecule were found”.

When we changed and corrected the manuscript according to all reviewers’ comments, we made a mistake by including the term “cardiogenic shock” in the section exclusion criteria. But as pointed out by the reviewer we have included “6% of cardiogenic shock”. Thus, we have corrected the mistake by withdrawing cardiogenic shock of exclusion criteria. We would like to apologize for this mistake.